# Digital TKA Alignment Training with a New Digital Simulation Tool (Knee-CAT) Improves Process Quality, Efficiency, and Confidence

**DOI:** 10.3390/jpm13020213

**Published:** 2023-01-26

**Authors:** Heiko Graichen, Marco Strauch, Michael T. Hirschmann, Roland Becker, Sébastien Lustig, Mark Clatworthy, Jacobus Daniel Jordaan, Kaushik Hazratwala, Rüdiger von Eisenhart-Rothe, Karlmeinrad Giesinger, Tilman Calliess

**Affiliations:** 1Department of Arthroplasty, General Orthopaedics and Sports Medicine, Asklepios Orthopaedic Hospital Lindenlohe, 92421 Schwandorf, Germany; 2Department of Orthopedic Surgery and Traumatology, Kantonsspital Baselland, University Hospital of Basel, 4101 Bruderholz, Switzerland; 3Department of Orthopaedic and Traumatology, Centre of Joint Arthroplasty West-Brandenburg, University of Brandenburg, 14770 Brandenburg, Germany; 4Department of Orthopaedic Surgery and Sports Medicine, FIFA Medical Center of Excellence, Croix-Rousse Hospital, Lyon University Hospital, 69004 Lyon, France; 5Faculté de Médecine Lyon Est, University Lyon, Claude Bernard Lyon 1 University, 69622 Lyon, France; 6Department of Orthopaedics, Ascot Hospital, Remuera, Auckland 3050, New Zealand; 7Division of Orthopaedic Surgery, Department of Surgical Sciences, Faculty of Medicine and Health Sciences Stellenbosch University, Tygerberg 7505, South Africa; 8Queensland Lower Limb Clinic, Mater Medical Center, Pimlico, QLD 4812, Australia; 9Department of Orthopedics and Sports Orthopedics, Klinikum Rechts der Isar, Technical University Munich, 81675 München, Germany; 10Department of Orthopaedics and Traumatology, Kantonsspital St. Gallen, 9000 St. Gallen, Switzerland; 11Articon Spezialpraxis für Gelenkchirurgie, Salem-Spital, 3013 Bern, Switzerland

**Keywords:** alignment, total knee arthroplasty, simulator, teaching tool, computer assistance, robotics

## Abstract

Individual alignment techniques have been introduced to restore patients’ unique anatomical variations during total knee arthroplasty. The transition from conventional mechanical alignment to individualised approaches, with the assistance of computer and/or robotic technologies, is challenging. The objective of this study was to develop a digital training platform with real patient data to educate and simulate various modern alignment philosophies. The aim was to evaluate the training effect of the tool by measuring the process quality and efficiency, as well as the post-training surgeon’s confidence with new alignment philosophies. Based on 1000 data sets, a web-based interactive TKA computer navigation simulator (Knee-CAT) was developed. Quantitative decisions on bone cuts were linked to the extension and flexion gap values. Eleven different alignment workflows were introduced. A fully automatic evaluation system for each workflow, with a comparison function for all workflows, was implemented to increase the learning effect. The results of 40 surgeons with different experience levels using the platform were assessed. Initial data were analysed regarding process quality and efficiency and compared after two training courses. Process quality measured by the percentage of correct decisions was increased by the two training courses from 45% to 87.5%. The main reasons for failure were wrong decisions on the joint line, tibia slope, femoral rotation, and gap balancing. Efficiency was obtained with a reduction in time spent per exercise from 4 min 28 s to 2 min 35 s (42%) after the training courses. All volunteers rated the training tool as helpful or extremely helpful for learning new alignment philosophies. Separating the learning experience from OR performance was mentioned as one of the main advantages. A novel digital simulation tool for the case-based learning of various alignment philosophies in TKA surgery was developed and introduced. The simulation tool, together with the training courses, improved surgeon confidence and their ability to learn new alignment techniques in a stress-free out-of-theatre environment and to become more time efficient in making correct alignment decisions.

## 1. Introduction

Total knee arthroplasty (TKA) is the treatment of choice for end-stage osteoarthritis (OA) of the knee and is frequently performed worldwide [1]. Full patient satisfaction is only achieved in 80% of cases, which is rather low compared to total hip arthroplasty [2,3,4,5]. One reason for this reduced rate might be the imprecision of conventional surgery. Over decades, TKA was performed with conventional instruments and, due to that, the direct linkage between bone cuts and gap widths was more experience than quantitative-data-based [6,7]. After the introduction of navigation and robotic systems, every step can be planned and executed quantitatively with high precision [8,9,10,11]. Although this did not automatically lead to improved patient satisfaction or patient outcome, the revision rates have been reduced at least in some studies and registries [12,13]. Clearly, this first disappointment in the value of such robotic systems makes one question whether the problem of limited patient satisfaction is not so much the consequence of missing technical precision but more of an unclear alignment and gap balance target. Using novel technology and still using a standardised approach with the same systematic targets in all different knee phenotypes might be a dead end.

Multiple authors have shown that a horizontal joint line, which is the alignment target in mechanical alignment, is present only in 20% of patients, and the mean medial proximal tibia angle (MPTA) is 87 degrees instead of 90 degrees [14,15,16]. Hazratwala et al. showed on more than 4000 pre-operative knees a wide variation in bony knee morphology, which is quantitatively normally distributed [17]. Hirschmann et al. demonstrated that various bony phenotypes (alignment phenotypes) exist, and Eller et al. recently yielded that the same is also true for the ligamentous situation [18,19]. In a recent paper, Graichen et al. presented that adjusted mechanical alignment reached the classical goals of leg alignment and balanced joints at a very high percentage, however, at the price of non-anatomical cuts [20]. This supports the philosophy that the classical goals might be appropriate in some, but not all, patients and that a patient-specific alignment strategy might offer a potential solution.

Multiple parameters of the bone (alignment phenotypes), as well as of the ligaments (laxity phenotypes), in extension, as well as flexion, are relevant to achieve a personalised TKA alignment. This multi-dimensional consideration needs to be performed before, in the beginning, and during TKA. However, this adds complexity and converts a rather simple mechanical alignment surgery into a more demanding planning process and TKA surgery. Therefore, the transition from one systematic standard alignment workflow to one of the existing individual workflows needs guidance and education. In addition, this transition in the OR is time-consuming and increases surgeon stress levels and consequent surgical errors [21]. It is likely that the increased complexity of computer-assisted surgery (CAS) is one of the main reasons why it gained only limited acceptance in the orthopaedic surgeon community [22,23,24,25,26,27].

This academic teaching challenge can be addressed in different ways. For example, with VR/AR solutions or by surgical simulators. As simulators have the advantage to allow training outside the OR, we decided to develop this TKA simulation tool. It is a simulated computer software that allows a surgeon to learn and practice one of many individualised alignment techniques in a simulated environment based on real case data. It offers training scenarios with various workflows, directly showing the differences between different alignment techniques. It enables surgeons to more easily understand the differences between the various workflows and to analyse whether this effect is similar in variable knee phenotypes.

The aims of this study were (1) to introduce a digital learning tool (TKA simulator), including various individual workflows, and (2) to evaluate its use on the surgeon’s workflow performance, efficiency, and confidence.

## 2. Materials and Methods

In the first step, we developed the interactive surface of this simulator. The name given to the simulator was Knee-CAT (CAT for Computational Alignment Trainer). In Knee-CAT, all surgical workflows and quantitative decisions of TKA surgery were implemented. The software demonstrates the cumulative and linked effects of bone resection on alignment and joint gaps in flexion and extension.

For the tibia implant positioning, one category for slope, another one for varus/valgus cut, and a third one for the resection height were implemented. The reference system is the mechanical axis of the tibia, obtained by palpating the 2 malleoli and the centre of the tibia head (ACL insertion). All decisions could be modified in steps of 0.1 mm or 0.1 degrees, and all were linked to each other and to the gap sizes, according to the typical workflow of navigation or robotic systems. The centre of rotation was placed in the centre of the tibia surface (for both directions, varus/valgus, and anterior–posterior slope). Therefore, an increase in tibia slope simultaneously affected the extension and flexion gap, however, in opposite directions. For the varus/valgus cut, all changes were performed again around this centre point. Hence, every degree of bone cut led to a 0.5-degree change on the medial and opposite on the lateral side. Medial and lateral cartilage loss was quantified in the patient briefing. A size of 2 mm described intact cartilage, and every loss was displayed in 0.5 mm steps. A size of 0 mm meant complete cartilage loss. The same was performed for the distal and posterior femur. Negative values described bone loss. In a separate step, the height of the tibia cut needed to be adapted to the desired height. Both these effects were again linked to extension and flexion gap sizes (Figure 1).

For the femur, seven categories of decision—one for distal varus/valgus cut angle, one for the amount of distal bone cut (medial and lateral femoral condyles), one for femoral rotation (relative to the posterior condyles), one for posterior condyle resections—were integrated. The reference is again the mechanical axis of the femur. These posterior condyle resections can be modified by three methods. Changing the femoral component size, altering the AP positioning of the femoral component, or changing the femoral flexion. Flexing the femoral component will reduce the flexion gap, while extending the femoral component will increase the flexion gap. This program is anterior-referenced, which is similar to some but not all navigation/robotic systems. Varus/valgus positioning was performed around the most prominent condyle, in cases where both condyles were equal, around the medial one. The femoral component rotation was performed around a central rotation point so that similar increasing and decreasing effects were seen for the medial and lateral posterior condyle resections (Figure 2). All reference points were highlighted by blue dots in the femoral model. With this constant centre pivot still, all options for implant positioning are possible; however, it sometimes requires a combination of changes in two planes to achieve the desired resection plane in terms of angle and height.

The simulations of the osteophyte removal and soft tissue release steps were based on analysing our 1000 patient data sets and the literature [28,29,30,31]. To simulate this part of the surgery, six categories for Varus and six for Valgus release, each with standardised values depending on the number of releases performed, were implemented, ranging from no release (just approach) up to maximal release (e.g., including the release of the superficial medial collateral ligament (sMCL) in varus knees). For each step of release, the corresponding effect on the medial and lateral extension and flexion gap was simulated (Figure 3).

After the surface was programmed according to the above-listed criteria, data sets of 125 previously operated knees were uploaded. Each file included anonymised basic patient information such as age, weight, height, and gender. All relevant knee parameters were included. These parameters were collected from patients’ radiographs (long-leg standing radiographs) as well as from intraoperative CAS data. The radiographic data included relevant bony angles such as the hip–knee–ankle angle (HKA angle), medial proximal tibia angle (MPTA), and mechanical lateral distal femoral angle (mLDFA), as well as a description of the cartilage situation according to the Kellgren–Lawrence classification. All six joint surfaces (medial and lateral tibia, distal and posterior condyles (medial and lateral)) were quantified with regard to their cartilage situation. A cartilage layer was simulated with 2 mm if the cartilage was intact and with 0 mm if completely worn out. Everything in between was measured intraoperatively and quantified in 0.5 mm increments. In cases of bone defects, negative values were stored. Clinically relevant information on the amount of fixed flexion deformity (FFD) or hyperextension was added to the system. Finally, gap data from navigation (Knee3, Brainlab, Munich-Germany) of all patients at the beginning of surgery was added and integrated into the simulator. The amount of gap data was reduced to the four relevant parameters of medial and lateral extension (0°) gap and medial and lateral flexion gap (90°).

Knee-CAT was programmed to simulate the following alignment workflows: Tibia-first workflows:

Mechanical alignment gap balanced (MA-TF), adjusted mechanical alignment (aMA-TF), constitutional varus (CV), anatomical alignment (AA), patient-specific alignment (PSA), inverse kinematic alignment (iKA), and Functional alignment (FA).

Femur-first workflows:

Mechanical alignment-measured resection (MA-FF) and kinematic alignment (KA), as well as restricted kinematic alignment (rKA). We integrated all of these workflows according to the definitions described in the literature [14,32,33,34,35,36,37,38,39,40,41,42,43,44,45,46]. Additionally, internationally known alignment specialists crosschecked their preferred alignment workflows concerning all definitions and their practical use (M. Hirschmann for aMA; S. Lustig for iKA and FA; K. Hazratwala for FA; K. Giesinger for CV; M. Strauch for AA; M. Clatworthy for PSA; and T. Callies for rKA and KA).

All 125 exercises (TKA surgeries) can be simulated with all alignment philosophies. Additionally, different balancing goals were integrated. Traditionally, the goal of gap balancing was equal gap size for all four gaps [40,41]. As a second option, a larger flexion gap of 2 mm compared with the extension gap was defined. In this scenario, medial and lateral gaps should be equal, both in extension and flexion. The third balancing goal option integrated into the tool is to leave the lateral flexion gap 2 mm larger than the medial flexion gap. Both extension gaps must equal the medial flexion gap. This represents the balancing goal defined in KA/RKA/FA and PSA. As all cases can be performed with all alignment philosophies and with all balancing goals, a final sample size of approximately 4000 simulation scenarios has been created so far. More cases are being added to the tool in order to include all phenotypes in larger numbers. With this data, future analysis can be performed.

To visualise the effect of different alignment techniques and balancing goals on bone cuts and gap values, a comparison function was developed and implemented. This function allows for comparing all decisions systematically to better understand the differences between the different workflows for each decision (Figure 4). The comparison function is only activated after the exercises with the different workflows are performed.

At the moment, 11 different workflows have been incorporated in Knee-CAT. Based on all definitions, a fully automatic evaluation system was developed (Figure 5).

This allows for quantifying each decision and determining the difference between the algorithm value and the value chosen by the trainee. Four categories for final evaluation were implemented: 1—bony cuts; 2—gap balancing; 3—HKA; and 4—time. The overall rating system is visualised in three colours, in a traffic light logic. Green means the decision was correct. The criteria to be rated as correct was defined to be within 1 mm or 1 degree from the algorithm-based decision. Between 1 and 2 mm, the colour of the box became orange, and every deviation larger than 2 mm or 2 degrees was visualised in red. An additional colour of blue was installed. This was displayed if the balancing goal could not be achieved with the defined alignment workflow, even if all bony cuts had been performed perfectly according to the algorithm definitions.

To train surgeons also on efficiency, the category of time was additionally integrated. All subunits of the exercise were analysed according to predefined times (tibia 3 min, soft tissue 2 min, and femur 5 min). If the trainee needed more than 10 min overall it turned orange, and if more than 12 min, it turned red.

In this study, the data from five training courses with different levels of surgeon experience were analysed. They were categorised into three groups: group 1, less experience (equals resident level); group 2, mid experience (equals senior surgeon less than 2 years); and group 3, extensive experience (more than 2 years of experience as a senior surgeon and a yearly number of more than 50 TKA). The overall number of trainees included was 40. Each evaluation was divided into two parts. The first evaluation was performed at the beginning of the basic course. After an introduction to the software and basic training on tool handling, five cases with mechanical alignment needed to be performed by the trainees as a homework exercise. The second evaluation was performed after the trainees had participated in an advanced alignment course. In this course, all workflow principles and case-based training of specific workflows were taught. Depending on the focus of the group, it was specific for PSA/FA and/or RKA/KA. In each training session, at least one simulated training case was performed together (trainer and trainees) in the group (online or face-to-face). After the advanced alignment course, the trainees received homework and performed 5–10 homework exercises. The results of this homework were used as a database for the second evaluation. The evaluation of each case and each trainee was based on the four categories described above, and the percentage of passed exercises was calculated. A subgroup analysis was performed for all four parameters of bone cuts, balancing, alignment, and time in order to evaluate the overall outcome. This was compared with the results at the beginning of training. Additionally, the most frequently made mistakes were listed, and finally, the effect of training on efficiency (time) was measured.

All trainees received an anonymised questionnaire giving them the opportunity to answer pre-defined questions on training quality, tool performance, and potential for improvement.

## 3. Results

### 3.1. Performance

At the beginning of each course, the majority of trainees failed to pass the exercises independent of the experience level of the trainees or the level of difficulty. The overall pass rate (green and orange) was only 45%. The two categories with the most problems were bone cuts (e.g., distal femoral cut) and balancing (Figure 6).

After the course training, the rate of successfully performed exercises increased significantly to an overall pass rate of 87.5% (77.5% green, 10% orange, and 12.5% red). This again was independent of the surgeon’s experience level. The category in which most failures occurred after training was bone cuts (e.g., distal femoral cut), while balancing and alignment were achieved in green in more than 95% of cases, and time was not a reason for failure in any of the cases (Figure 6).

### 3.2. Efficiency

At the beginning of the training courses, the average time per exercise was 4 min and 28 s, with a range from 2 min 4 s up to 8 min 45 s. After the training course, the average time per exercise was shortened to 2 min 35 s, ranging from 1 min 44 s to 4 min 2 s. This was an overall reduction in time of 42%.

### 3.3. Confidence

All 40 trainees returned the evaluation sheet, and all categories were filled out and could be analysed. A total of 100% of trainees rated the tool as helpful or extremely helpful to learn CAS and robotic TKA surgery and to learn new alignment philosophies. All experienced the tool as a huge step forward compared to classical training in the OR.

A total of 95% rated the training and the integrated textbook as an important part of the learning experience. However, 85% described that using the tool was not self-explanatory and that additional training courses and feedback sessions were very important for general understanding and improvement.

The overall experience of the tool was rated good/very good, and 92.5% of trainees recommended the use of the tool for resident, registrar, and/or fellow training. The only limitation that was mentioned by more than 20% of trainees was the monitor surface. The integration of a 3D model, closer to existing navigation or robotic screens, was recommended to make the learning experience closer to their daily practice.

As for the next steps for the future development of the tool, the integration of X-rays with fully automatic measurement of bony anatomical angles was emphasised by three of the trainees.

## 4. Discussion

In this study, a digital training platform for simulating TKA surgery with various alignment workflows was introduced. The analysis of the first five training courses showed that all participants had a significant improvement in process quality and efficiency. The number of mistakes was reduced, and the overall time was 40% shorter than at the beginning of the training. It provides improved education for surgeons outside the OR and before using new digital tools such as robots. Further, it eased the transition towards more individual alignment techniques. Based on specific, real patient data, typical surgical scenarios were simulated, and with the help of specific algorithms, a fully automatic evaluation of each exercise was displayed directly after the case was finished. The integrated comparison function, for the different workflows and alignment techniques, helped to improve the understanding.

Traditionally, the teaching and training of TKA surgery was performed in the OR and focused on the optimised handling of conventional instruments. The evaluation of surgical success was mainly based on qualitative parameters, summarised as the surgeon’s experience. This made some aspects of the decision-making process difficult to teach to younger colleagues, and a prolonged learning curve was the consequence. With the introduction of CAS and robotics in TKA surgery, the surgical process became quantitative. This allows one to measure surgical quality by analysing those decisions. However, in the beginning, these parameters are very complex to interpret, and a specific learning curve for these digital workflows is the consequence. If such parameters are understood, the knowledge transfer of the different surgical steps to younger surgeons becomes a lot more transparent and reproducible [26,47].

Recently, various digital options for the training of TKA surgery, such as virtual reality (VR) and augmented reality (AR), have been introduced. Various authors have pointed out the strength of AR technology, as it allows superimposing real radiological images or templates onto the real operating field. This additional information can ease the orientation for bone cuts. The difference compared to our simulation tool is that all AR training still needs to be performed during surgery. Another challenge of AR tools is the learning curve of handling them. Both problems make the use, at least in the beginning, stressful and time-consuming. On the other hand, after the learning curve has been overcome, it may help surgeons with quantitative decisions in real cases [23,48].

In other fields of surgery/orthopaedics such as arthroscopy or general surgery, more classical teaching tools, such as OR simulators, have a long history. The objective of these tools is to train specific technical skills, in particular, how to use instruments more efficiently. Knee-CAT was developed with the purpose of creating an “OR-similar” environment and allowing the teaching of digital TKA decisions and also training on alignment techniques. This training is performed before surgery, and as in other simulators, outside the OR. In most of these simulators, a standard anatomic situation is simulated as real as possible, preparing the surgeon for real-life surgery [49]. The simulation of specific pathologies, however, is limited, which is a relevant difference to Knee-CAT offering the simulation of all kinds of deformities and pathologies for training. The advantage of all simulations is that various important surgical steps of the procedure can be trained specifically without harming the patient and without stress in the OR. The newly introduced TKA simulator combines both advantages, and on the one hand, every quantitative decision that needs to be made can be trained before a real TKA is performed. This simulator offers additionally a fully automatic evaluation based on an integrated algorithm. This algorithm includes the definitions of all workflows described in the literature. Based on the deviations from optimum, the results are displayed in traffic light colours. On the other hand, different knee pathologies and/or phenotypes can be simulated, and by that, a deepened experience and knowledge can be transmitted. Like all the other simulators, so far, this TKA simulator is a pure training tool; therefore, it cannot be used in the OR as a decision-supporting tool. At the moment, different workflows can be compared with the tool. A scoring system, which assesses workflows regarding their quality, is not included. However, it can be one future development direction of the tool.

In TKA, various alignment techniques have been introduced, and the steps for each workflow have been described in the literature. As multiple quantitative decisions need to be made in TKA surgery, and all these decisions are linked directly to the gap values and to the following steps, it is complex to start the transition to these new workflows. Starting this learning experience in the OR is stressful and failure-prone. Additionally, it is time-consuming, which is potentially directly related to an increased infection risk. Therefore, separating alignment training from OR execution is beneficial, and this aspect was confirmed by the given feedback from the trainees.

Comparing the effect of different workflows on alignment and balance as on bony resections at the different joint planes was rated by the trainees as another very important feature of the tool. This might allow, in the future, more detailed knee phenotypes analysis compared with the current pure coronal bone analysis [19]. As gap data of different flexion angles are also included, additional phenotype analysis in flexion can be performed.

The tool, however, still has some limitations. The main limitation is that osteophyte removal and soft tissue management are simulated based on standard values obtained from in vivo data and from biomechanical simulation data [15,28,29,30,31]. This is a simplification; however, the general effect can be simulated. The integration of individual patient values can only be performed intraoperatively. With this data, the tool can become part of the intraOP workflow and might move away from a poor education tool towards a medical device with all regulatory challenges. Another limitation is that all patient data implemented in the tool so far are from Caucasian knees, which might affect the results of the different workflows. As another limitation, the monitor appearance was mentioned by some trainees. Therefore, in the new software version, demonstrated here in all illustrations, a 3D model of the femur and tibia in different views was implemented.

## 5. Conclusions

In this study, Knee-CAT—a novel TKA navigation simulation tool for all quantitative decisions and all modern alignment techniques—was introduced. It allows the efficient training of all OR decision steps before surgery, and by that, the learning curve is separated from surgery, meaning that experience can be collected before the first real patient is operated on. This increases patient safety and process efficiency. The experience of the first training courses proved this concept.

## Figures and Tables

**Figure 1 jpm-13-00213-f001:**
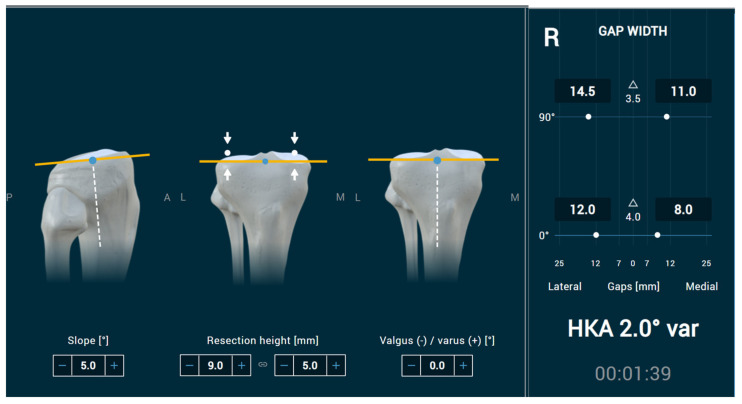
Tibia decisions of slope, resection height, and varus/valgus cut, with all referenced to the centre point (blue dot) of the tibia and both directly linked to the extension/flexion gap size.

**Figure 2 jpm-13-00213-f002:**
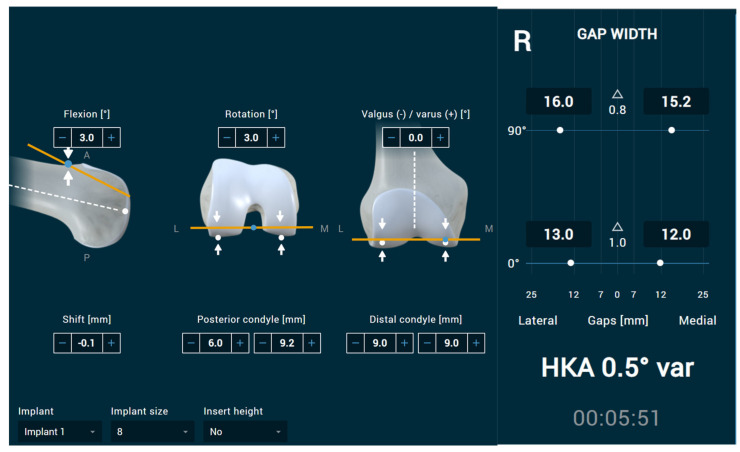
User interface of the femoral planning/decision monitor. All reference points are shown as blue dots. All parameters on the left side are directly linked to extension and flexion gap size.

**Figure 3 jpm-13-00213-f003:**
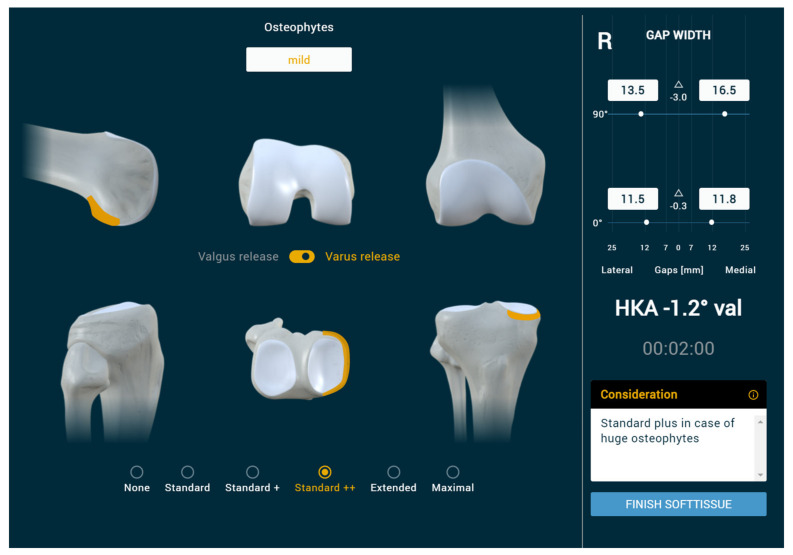
Different release steps are listed at the bottom and can be chosen based on gap differences or workflow. This can be performed for varus and valgus knees. The anatomical region of release/osteophyte removal is demonstrated depending on the category from none (just approach) to maximal (including the superficial MCL). In this case, std ++ was chosen, demonstrating the removal of osteophytes all around the medial tibia and the posterior femoral condyles.

**Figure 4 jpm-13-00213-f004:**
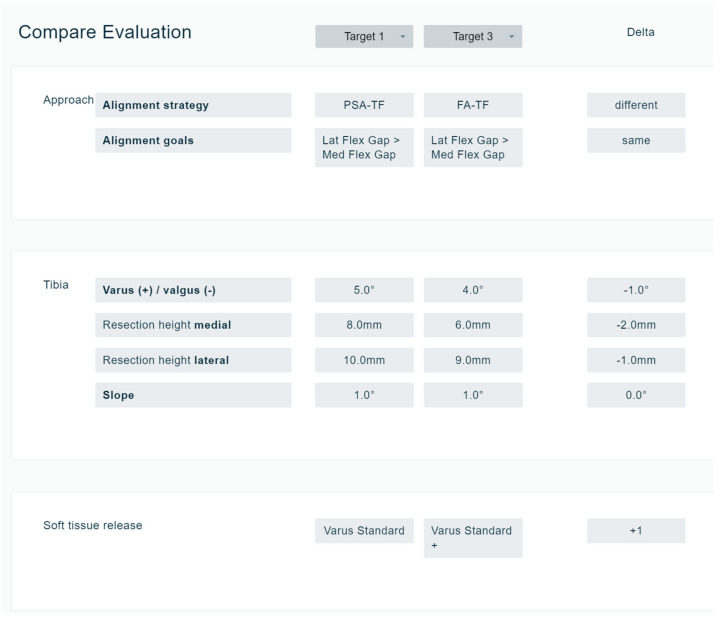
Comparison function. In all cases, all decisions of different alignment philosophies can be seen on one screen (in this figure, only for tibia and soft tissue). This allows a comparison of different philosophies in different cases. Here, the difference between PSA-TF and FA-TF is shown for tibia and soft tissue decisions.

**Figure 5 jpm-13-00213-f005:**
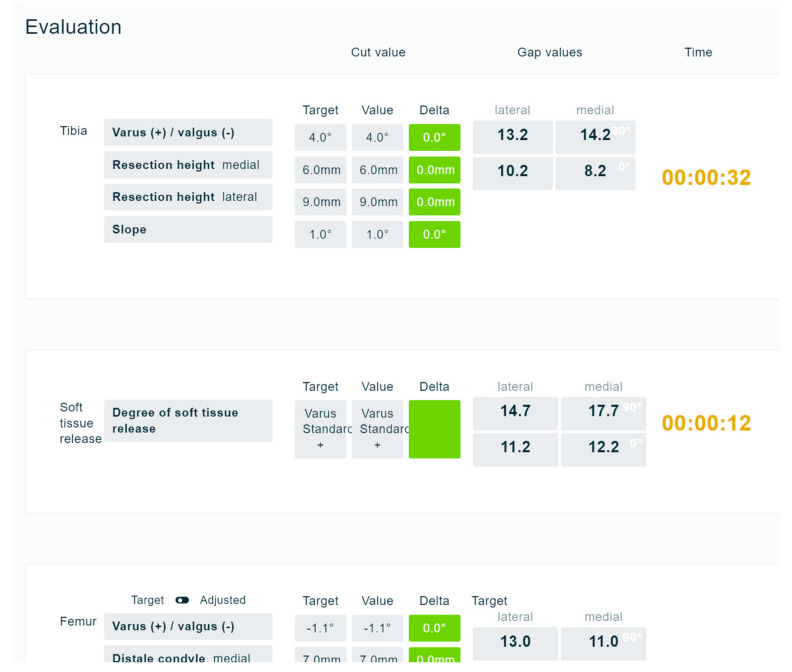
Part of the evaluation screen. Each decision made by the trainee is shown in the column “value” and is directly compared with the column “target”. Target values are representing the optimal, algorithm-based values. All deltas are quantified and displayed in a traffic light system. If differences are minimal (<1 mm or <1°), in green; if maximal (>2 mm or 2°), in red; and everything in between 1 and 2 mm or 1 and 2°, in orange.

**Figure 6 jpm-13-00213-f006:**
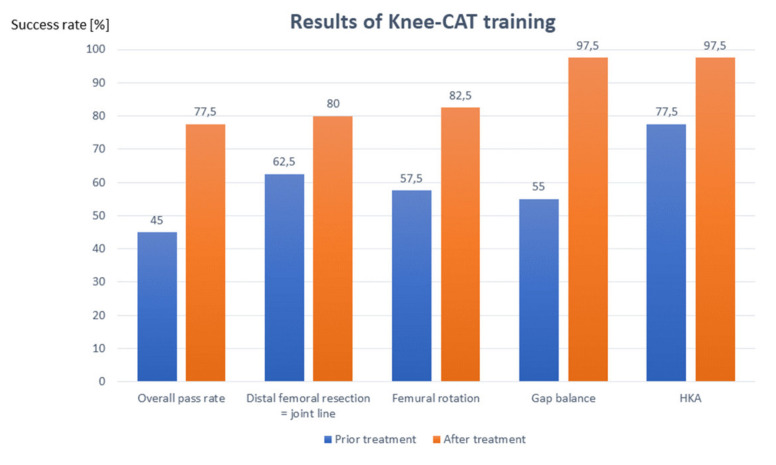
Summary of the results showing the overall improvement after Knee-CAT training. The most common problems were the distal femoral cut, femoral rotation, and gap balancing. Each category was again improved after training, with gap balance and HKA to almost 100%.

## Data Availability

Data was collected and stored according to the local requirements but is unavailable due to privacy or ethical restrictions.

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
