# Peer review of "Digital TKA Alignment Training with a New Digital Simulation Tool (Knee-CAT) Improves Process Quality, Efficiency, and Confidence"

_jpm, 2023, doi:10.3390/jpm13020213_

Round 1

Reviewer 1 Report

Dear Authors,

The manuscript is well written. Below are some minor considerations for your consideration.

Abstract

Line 36-7: Please rephrase for clarity.

Introduction

The introduction provides a thorough overview of the rationale from a clinical perspective, but fails to introduce the novelty of the study in terms of an alternative training tool. Consider adding elements form the discussion (line 301-43)

Methods

line 145: Consider replacing "according" with "corresponding".

line 157: Consider replacing "radiological" with "radiographic".

line 167: Delete "also".

line 154: Consider adding basic patient characteristics, e.g. age, sex, BMI.

line 174 and others: For abbreviations, please consider changing the first letter to lower case: i.e. aMA, iKA, rKA instead of AMA, IKA and RKA. 

line 176-7: Please rewrite into a full sentence.

line 197-8: Please rephrase for clarity.

Results

line 252-61: Please be consistent in the way the results are presented. For example: "...(green and orange) was only 45%." (line 254) versus "...77.5% green, 10% orange and 12.5% red." line 257.

line 267-71: Please change the order. Start with initial mean and range, then final mean and range, and then the improvement.

Discussion

1st paragraph: The authors could consider changing the order. In its current form the discussion starts with (vague) qualitative findings and ends with more (substantial) quantitative findings. By starting with the quantitative findings, the first paragraph could be more punchy and provide justification  for "improved education" and "eased transition".  

Author Response

Abstract Line 36-7 Please rephrase for clarity
Has been modified accordingly
Initial data was analysed regarding process quality and efficiency and compared after two training courses.

Introduction The introduction provides…
We have added this aspect accordingly (line 91-93)
This academic teaching challenge can be addressed in different ways. For example with VR/AR solutions or by surgical simulators. As simulators have the advantage to allow training outside the OR we decided to develop this TKA simulation tool.

Methods

Line 145 (now 152) corresponding was used

Line 157 (now 164) changed to radiographic

Line 167 (now 173) also eliminated

Line 154 (now 161-162) included accordingly
Each file included anonymised basic patient information such as age, weight, height, and gender

Line 174 and further (now 180 and further) changed accordingly
Mechanical alignment-gap balanced (MA-TF), adjusted mechanical alignment (aMA-TF), Constitutional varus (CV); Anatomical alignment (AA), Patient specific alignment (PSA), Inverse kinematic alignment (iKA) and Functional alignment (FA).

Femur first workflows:

Mechanical alignment-measured resection (MA-FF) and kinematic alignment (KA) as well as restricted kinematic alignment (rKA). We integrated all these workflows according to the definitions described in the literature [14, 32-46]. Additionally, internationally known alignment specialists crosschecked their preferred alignment workflows concerning all definitions and its practical use (M. Hirschmann for aMA; S. Lustig for iKA and FA, K. Hazratwala for FA, K. Giesinger for CV, M. Strauch for AA; M. Clatworthy for PSA and T. Callies for rKA and KA).

Line 176-7 List of alignment workflows was rephrased

Line 197-98 (now 200-202) has been rephrased
More cases are being added into the tool in order to include all phenotypes in a larger number. With this data future analysis can be performed.

Results

Line 252-61: Please be consistent… (now 265…) changed accordingly
After the course training the rate of successful performed exercises increased signifi-cantly to an overall pass rate of 87.5% (77.5% green, 10% orange and 12.5% red). This again was independent of the surgeon’s experience level. The category in which most failures occurred after training was bone cuts (e.g. distal femoral cut), while balancing and alignment was achieved in green in more than 95% of cases and time was not a reason for failure in any of the cases (Fig. 6).

Line 267-71: Please change the order… (now 277-279) Changed accordingly
After the training course, the average time per exercise was shortened to 2 minutes 35 seconds, ranging from 1 minute 44 seconds to 4 minutes 02 seconds.

Discusssion

1st paragraph:… (lines 299-303) changed accordingly
In this study, a digital training platform for simulating TKA surgery with various alignment workflows has been introduced. The analysis of the first five training courses showed that all participants had a significant improvement in process quality and efficiency. The numbers of mistakes were reduced, and the overall time was 40% shorter than at the beginning of the training.

Reviewer 2 Report

Overall an excellent study which could have high utility in Resident training. Even experienced surgeons are likely to receive benefits from KNEE-CAT.  

Line 25-26. ‘Individual alignment philosophies have been introduced to restore patients’ unique anatomical variations during total knee arthroplasty’

In my view, the word ‘philosophies’ is not a good word to use in a scientific paper. A better word could be ‘techniques’ or ‘modalities’, something like that.

The word ‘philosophies’ in the context of the paper, sounds like ‘opinion without any sound scientific data’.  

Line 375. ‘Funding: This research received no external funding.’

This is a formidable study which must have taken a great deal of time and manpower. The funding must have been substantial. Moreover, several different institutions were involved. Was it funded by a European Grant for a multi-Center Study? By a company? By venture funding? It will be instructive to know more details. Whatever the funding source, this would not detract from the quality of the work.  

Line 107-108. ‘For the tibia implant positioning, one category for slope, another one for varus/valgus cut and a third one for the resection height were implemented.’

It should be stated what was the reference axes of the tibia, to which the angles are referred. Is this the anatomic axis of the tibia ie from center of proximal tibia, to center of the malleoli? Also, there is no mention here of how to deal with the osteoarthritic reality where there is asymmetric cartilage loss, as well as bone collapse.

The same question for the femur side.  

Line 145-146. ‘For each step of release the according effect on 145 the medial and lateral extension and flexion gap was simulated (Fig. 3).’

How were the releases of ligaments translated into changes in angulation or gap? Eg the MCL can be made less stiff by removal of part of the ligament, but its overall length may be the same. Do the authors in any way take account of an estimate of ligament stiffness properties ie Newtons/mm ?  

General comment at this stage: In a 2017 J Arthroplasty study: A Targeted Approach to Ligament Balancing Using Kinetic Sensors Kenneth A. Gustke, MD a , Gregory J. Golladay, MD b , Martin W. Roche, MD c , Leah C. Elson, BSc d , Christopher R. Anderson, MBA, MSc d, multiple balancing scenarios involving many soft tissue structures were described. These releases were decided on measurements made at the time of surgery and could not be predicted ahead of time.  I realise that the Training Model could not necessarily include all of this detail, but at least it should be addressed in the paper in some way.  

Discussion: I am not sure if the paper clearly explains what is the purpose of KNEE-CAT, and what is the purpose of the training itself. Does each surgeon simulate each different bone cut/soft tissue modality ? How is the score obtained? Ie based on what parameters? Does KNEE-CAT indicate at all which might be the best alignment to use for a particular patient? And so on.

Author Response

Overall an excellent study…
Thank you a lot for this comment

Line 25-26 Individual alignment philosophies…
The term philosophies has been changed throughout the manuscript to techniques

Line 375 (now 392-398) Funding
Additional information was included
Funding: This research received no external funding. Indeed, a lot of time from various people has been invested. Knee-CAT is the product of SmartOrthoSolutions. Heiko Graichen (first author) is Medical Head and one of the co-founders. All 4 founders invested in this project and the company. No external funding from other companies was used for Knee-CAT development. Meanwhile Knee-CAT can be used in different set-ups (Scientific Societies, Conferences and Company sponsored education). Meanwhile the first licenses have been sold, however, the published development of the tool was without external funding, just sponsored by the owners.

Line 107-108 (now 109-112) For the tibia implant position…
Thank you for this comment. We have added the specific information for tibia and femur.
For the tibia implant positioning, one category for slope, another one for varus/valgus cut and a third one for the resection height were implemented. The reference system is the mechanical axis of the tibia, obtained by palpating the 2 malleoli and the centre of the tibia head (ACL insertion).

Line 145-146 (now ) For each step of release…
The stiffness of the ligaments was not measured. Lax ligaments display large gaps and stiff ligaments smaller gaps accordingly.

General comment at this stage: …
This is an interesting additional approach. Wi will include these findings in our future investigation.

Discussion: I am not sure… (now lines 333-334, 349-152)
We have added additional information accordingly. At this moment, KneeCat cannot provide the surgeon with the best workflow information.
Knee-CAT is developed with the purpose of creating an “OR-similar” environment and allow teaching of digital TKA decisions and also training on alignment techniques.
At the moment, different workflows can be compared with the tool. A scoring system which assess workflows regarding their quality is not included. However, it can be one future development direction of the tool.